# Structural basis for recognition of 53BP1 tandem Tudor domain by TIRR

Yaxin Dai[1,2], Aili Zhang[3], Shan Shan[1], Zihua Gong[3] & Zheng Zhou[1,2]

P53-binding protein 1 (53BP1) regulates the double-strand break (DSB) repair pathway choice. A recently identified 53BP1-binding protein Tudor-interacting repair regulator (TIRR) modulates the access of 53BP1 to DSBs by masking the H4K20me2 binding surface on 53BP1, but the underlying mechanism remains unclear. Here we report the 1.76-Å crystal structure of TIRR in complex with 53BP1 tandem Tudor domain. We demonstrate that the N-terminal region (residues 10–24) and the L8-loop of TIRR interact with 53BP1 Tudor through three loops (L1, L3, and L1'). TIRR recognition blocks H4K20me2 binding to 53BP1 Tudor and modulates 53BP1 functions in vivo. Structure comparisons identify a TIRR histidine (H106) that is absent from the TIRR homolog Nudt16, but essential for 53BP1 Tudor binding. Remarkably, mutations mimicking TIRR binding modules restore the disrupted binding of Nudt16-53BP1 Tudor. Our studies elucidate the mechanism by which TIRR recognizes 53BP1 Tudor and functions as a cellular inhibitor of the histone methyl-lysine readers.

[1] National Laboratory of Biomacromolecules, CAS Center for Excellence in Biomacromolecules, Institute of Biophysics, Chinese Academy of Sciences, Beijing 100101, China. [2] University of Chinese Academy of Sciences, Beijing 100049, China. [3] Department of Cancer Biology, Cleveland Clinic Lerner Research Institute, Cleveland 44195 OH, USA. These authors contributed equally: Yaxin Dai, Aili Zhang, Shan Shan. Correspondence and requests for materials should be addressed to Z.G. (email: gongz@ccf.org) or to Z.Z. (email: zhouzh@sun5.ibp.ac.cn)

I n eukaryotic cells, the repair of the double-stranded DNA breaks (DSBs) is achieved by two mechanistically distinct pathways: non-homologous end-joining (NHEJ) and homologous recombination (HR)[1,2]. The fate decision of DSB repair process is of paramount importance for ensuring genome stability and is subject to a precise regulation during different phases of the cell cycle[3]. The tumor suppressor p53-binding protein 1 (53BP1) plays a pivotal role in orchestrating the choice of DSB repair pathway. The 53BP1 promotes NHEJ-mediated DSB repair that keeps DSB ends from resection and prevents HR by counteracting the function of breast cancer-associated gene 1 (BRCA1) in the HR pathway[4–8]. Cells lacking BRCA1 are defective in HR-mediated DSB repair and extremely susceptible to treatment with PARP inhibitors (PARPi)[9,10]. However, loss of 53BP1 in BRCA1-deficient cells restores the HR repair and alleviates the cell sensitivity to PARP inhibition[6,7].

Central to the 53BP1 function is the recruitment of 53BP1 to the damaged chromatin via the recognition of di-methylated lysine 20 of histone H4 (H4K20me2) and ubiquitinated lysine 15 of histone H2A (H2AK15ub). Binding of these chromatin epitopes are mediated by the 53BP1 tandem Tudor domain and the ubiquitin-dependent recruitment (UDR) motif, respectively[11,12]. Recognition of H4K20me2 is conferred by 53BP1 residues W1495, Y1502, F1519, Y1523, and D1521, which form an aromatic cage structure within 53BP1 tandem Tudor domain[12]. Moreover, similar aromatic cages with substantial structure conservation have been discovered in many other histone methyl-lysine readers, underscoring the role of conserved recognition mode in methyl-lysine reading[13,14].

Early studies indicated that some indirect regulatory mechanisms such as masking H4K20me2 mark by L3MBTL1 and JMJD2A coupled with H4K16 acetylation restrict 53BP1 access to chromatin[15–17]. More recently, a Tudor-interacting repair regulator (TIRR) has been characterized as a novel 53BP1 regulator, which directly binds to 53BP1 Tudor domain and blocks its H4K20me2 binding surface[18,19]. While the TIRR-deficient cell displays increased sensitivity to persistent DSBs and ionizing radiation (IR), overexpression of TIRR compromises the formation of 53BP1 foci and reduces the sensitivity of BRCA1-mutated cells to PARPi[18,19]. Further analyses revealed that TIRR regulates 53BP1 activity at multiple levels and determines the selection of DSB repair pathway by interacting with the tandem Tudor domain of 53BP1. However, the underlying mechanism of this regulation process remains unclear.

In this study, we report the high-resolution structure of TIRR in complex with the tandem Tudor domain of 53BP1. In the crystal structure, the N-terminal region and the L8-loop of TIRR form an extensive binding interface with three loops of 53BP1 Tudor. TIRR, which exhibits robust binding with 53BP1 Tudor, masks the binding surface of H4K20me2 and regulates 53BP1 functions in vivo. A further study identifies TIRR residues, which are necessary and sufficient to confer the NUDIX hydrolases, Nudt16 an ability for 53BP1 Tudor binding[20]. Collectively, these findings reveal the structural basis for recognition of 53BP1 Tudor by TIRR and elucidate the mechanism by which TIRR functions as a bona fide cellular inhibitor of 53BP1.

## Results

**TIRR interacts with 53BP1 tandem Tudor domain.** Previous studies have shown direct binding of 53BP1 Tudor to TIRR. Other regions of 53BP1, like the N-terminal domain enriched with 28 × S/TQ, also plays a role in TIRR interaction[18]. To identify a minimal 53BP1 region that is sufficient to dictate TIRR binding (Fig. 1a), we performed pull-down assay using different 53BP1 fragments that comprise the tandem Tudor domain

(Tudor) and/or the UDR domain (Supplementary Fig. 1A). These results showed that both 53BP1 Tudor domain and Tudor-UDR domain could pull down TIRR (Supplementary Fig. 1A). Consistently, isothermal titration calorimetry (ITC) analyses showed that bindings of 53BP1 Tudor and Tudor-UDR to mouse TIRR yield the same binding association constant ($K_d = 0.9\ \mu M$) (Fig. 1b). These results suggested that 53BP1 Tudor is sufficient for robust TIRR binding. It is worth noting that the human TIRR, which shares 90% sequence similarity with its mouse homolog, displays a nearly identical binding to 53BP1 Tudor ($K_d = 0.9\ \mu M$) (Supplementary Fig. 1B), indicating a conserved binding mode adopted by TIRR from both species.

Since ITC data suggested a 1:1 binding stoichiometry for TIRR and 53BP1 Tudor (Fig. 1b), and TIRR displays a tendency to form a homodimer at the physiological condition (Supplementary Fig. 1C), we speculated that TIRR and 53BP1 Tudor should form a 2:2 stoichiometry tetramer complex, as such one TIRR dimer needs to interact with two 53BP1 Tudor monomers. The results of analytic ultracentrifugation analysis showed that TIRR/53BP1 Tudor complex assembled in equal molar ratio predominantly forms a 2:2 tetramer with a determined molecular weight of 67 kDa (4.1S) (sedimentation velocity) (Fig. 1c). In contrast, TIRR and 53BP1 Tudor premixed at 2:1 ratio results in a complex with substantially lower molecular weight. The molecular weights determined by analytic ultracentrifugation are 48 kDa (4.0S) (by sedimentation velocity) and 54 kDa (by sedimentation equilibrium), which are in agreement with the size of TIRR/53BP1 Tudor complex in 2:1 stoichiometry (Supplementary Fig. 1C). These findings strongly suggested that one TIRR homodimer possesses dual 53BP1 Tudor-binding site, as such they form complexes in either 2:2 or 2:1 stoichiometry.

We used combinations of TIRR from human and mouse to assemble TIRR/53BP1 Tudor complex in 2:2 or 2:1 stoichiometry for crystallography (Supplementary Fig. 2). The crystal structure of the complex reconstituted with mouse TIRR and 53BP1 Tudor at 2:1 molar ratio is determined at 1.76-Å resolution by molecular replacement. The unbiased electron density maps are clearly observed (Supplementary Fig. 3). The final model displayed good crystallographic and geometric statistics ($R_{work}/R_{free} = 0.20 / 0.22$) (Table 1). Following model building and refinement, one TIRR homodimer and one 53BP1 tandem Tudor domain are observable in each asymmetric unit, indicating a 2:1 stoichiometry for TIRR–53BP1 Tudor interaction in the crystal structure (Fig. 1d).

**Overall structure of the TIRR–53BP1 Tudor complex.** In the complex structure, a TIRR monomer comprises seven α-helices (α1–α7) and seven β-strands (β1–β7), whereas 53BP1 Tudor comprises two α-helices (α1–α2) and ten β-strands (β1–β5, β1′–β5′) (Fig. 1a). It is worth noting that the binding of one 53BP1 Tudor to TIRR dimer does not block the Tudor-binding site on the unbound TIRR protomer, suggesting that two molecules of 53BP1 could in theory bind to one TIRR dimer. Superimposing the complex structure with TIRR dimer structure[21] (PDB 4ZG0) or 53BP1 Tudor structure[12] (PDB 2G3R) resulted in a root-mean-square deviation (RMSD) value of 0.370 Å (for TIRR) and 0.470 Å (for 53BP1 Tudor) respectively, suggesting that both TIRR and 53BP1 Tudor retained structures similar to their free forms.

The complex structure shows that the N-terminal region (residues 10–24) and the L8-loop of TIRR are engaged with a concave surface of 53BP1 Tudor, which is delineated by three loops (L1, L3, and L1′) of 53BP1 Tudor (Fig. 1d). While the N-terminal region of TIRR engaged with the L1-loop and L3-loop of 53BP1 Tudor, TIRR L8-loop mainly interacted with the 53BP1

Tudor L1′-loop (Figs. 1d, 2). It is worth noting that residues distinct in human and mouse TIRRs either locate far from the TIRR/53BP1 Tudor binding interface or show no binding with 53BP1 Tudor (Supplementary Fig. 2).

**TIRR residues influence complex integrity.** Structure analysis reveals that binding of TIRR and 53BP1 Tudor are stabilized by multiple hydrophobic and polar interactions. TIRR residues I12, L20, and W24 in the N-terminal region, together with residue L101 in β5-strand, engage with 53BP1 residues W1495, Y1523 (Fig. 2a, b). These hydrophobic interactions are further stabilized by hydrogen bonds formed between TIRR residue K10 and 53BP1 residues W1495, Y1523, and between TIRR residue W24 and 53BP1 residue D1521 (Fig. 2b).

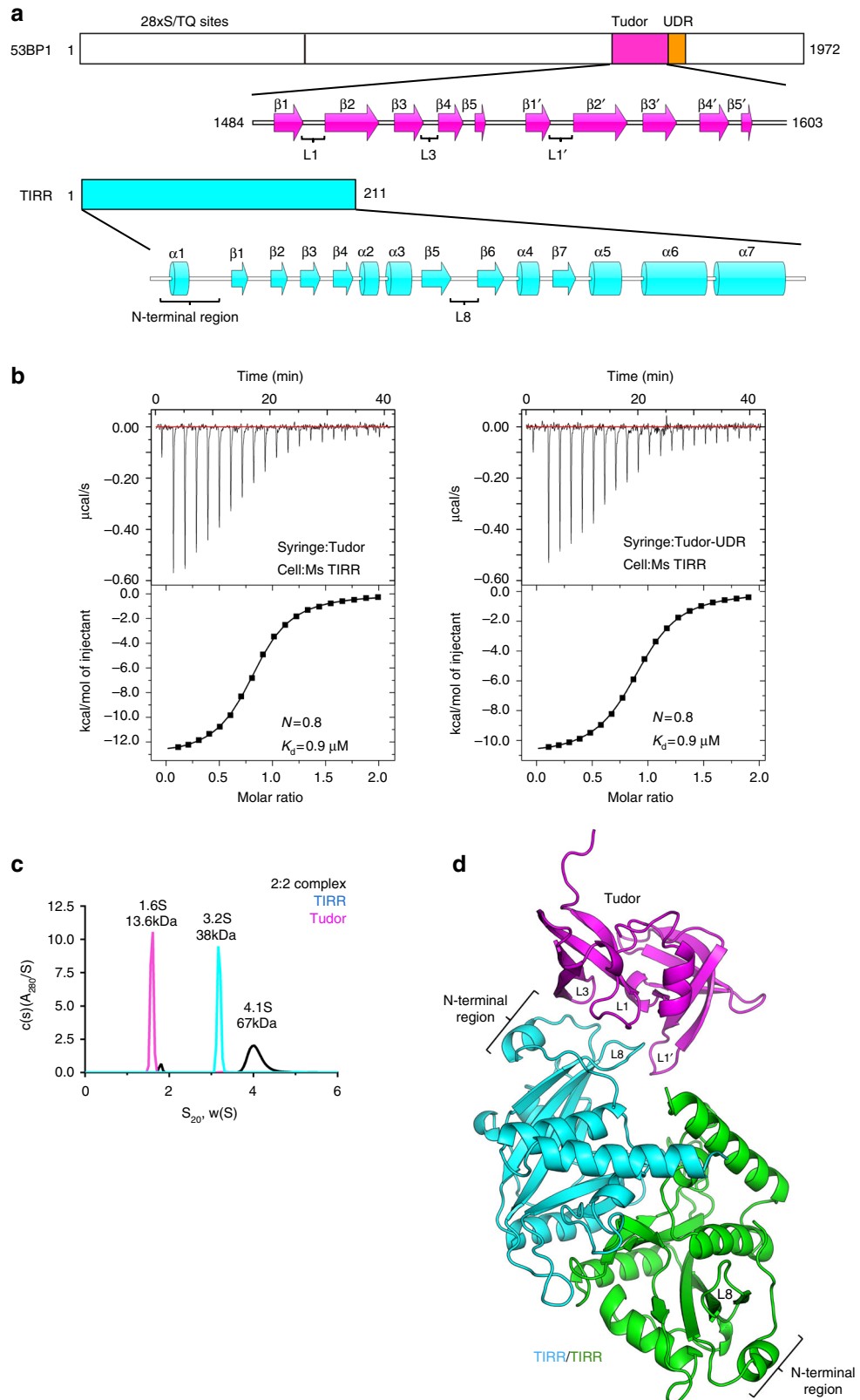

In addition to the intermolecular contacts between TIRR N-terminal region and the Tudor domain of 53BP1, hydrophobic and polar interactions have also been observed between TIRR L8-loop and 53BP1 Tudor. In brief, TIRR residues P105, H106 form hydrophobic cluster with 53BP1 residues Y1500, Y1502 in L1 loop and L1547, and F1553 in L1′ loop (Fig. 2c). The side chains of TIRR residues R107, W24 buried inside the hydrophobic cleft (Fig. 2d). Moreover, a hydrogen bond network is formed between TIRR residue P105, H106, R107, and 53BP1 residues Y1502, Y1552 (Fig. 2c) and D1521, S1503, and M1584 (Fig. 2d).

To identify TIRR residues essential for 53BP1 Tudor binding, we generated TIRR mutants each containing a single-residue mutation and used ITC to measure their binding affinities decreases. In comparison with the wild-type TIRR (wt), mutation of K10E in the N-terminal region resulted in a ~23-fold binding affinity decrease ($K_d$ of 20.4 μM) (Fig. 2e, Table 2), coinciding with the previous results showing that a TIRR K10E mutant failed to bind 53BP1 in vivo[18]. Remarkably, TIRR mutant I12G and L20A exhibited substantially reduced binding for 53BP1 Tudor by ~43-fold and ~28-fold, respectively (Fig. 2e, Table 2), and bindings of mutants L20G and W24A were completely abolished (Fig. 2e, Table 2). Examination of mutation located in TIRR L8-loop demonstrated that mutation of P105A and H106A displays moderate effect on 53BP1 Tudor binding by decreasing the affinity by ~5-fold ($K_d$ of 4.3 μM) and ~4-fold ($K_d$ of 3.9 μM), whereas mutation of R107A abolished the 53BP1 Tudor binding (Fig. 2e, Table 2). The center location and extensive 53BP1 interaction of TIRR residues W24 and R107 explain the drastic effects of their alanine mutations on 53BP1 Tudor binding and underscores the critical roles of both residues (Fig. 2d). Collectively, we conclude that the N-terminal region and L8-loop of TIRR are critical for 53BP1 Tudor recognition.

**Recognition of 53BP1 Tudor by TIRR blocks H4K20me2 binding.** It is reported that TIRR inhibits 53BP1 function by abrogating H4K20me2-53BP1 Tudor association[18]. Structural comparison reveals that while the 53BP1 Tudor binding mode for TIRR is distinct from that for H4K20me2, the binding interface of 53BP1 Tudor for TIRR and H4K20me2 are mutually exclusive (Fig. 3). We observed that H4K20me2-bound aromatic cage, which contains 53BP1 residues W1495, Y1523, D1521, Y1502, and F1519, undergoes conformational changes upon TIRR interaction (Fig. 3a). In particular, residues W1495 and Y1523 display the largest conformational change and completely alter the structure of the aromatic cage (Fig. 3a). Further analyses revealed that the conformation of H4K20me2-bound 53BP1 Tudor is incompatible with TIRR binding (Supplementary Fig. 4), suggesting a substantially altered binding surface caused by TIRR binding. Indeed, the major contact area of 53BP1 Tudor/TIRR, as calculated by PISA[22], spans 685.2 Å of the surface area of one Tudor molecule (binding energy of −8.9 kcal/mol). On the contrary, the contact area of 53BP1 Tudor/H4K20me2, calculated from the crystal structure of 53BP1 Tudor/H4K20me2 (PDB 2IG0), is 256.9 Å (binding energy of −3.2 kcal/mol) (Fig. 3b). These observations agree with our ITC results that demonstrate that 53BP1 Tudor binds to TIRR with a significantly higher

binding affinity than that of H4K20me2 (Supplementary Fig. 4). We conclude that TIRR disrupts the bindings of 53BP1 Tudor to H4K20me2 or other methylated-lysine by engagement with 53BP1 Tudor.

We next performed ITC to investigate the effect of 53BP1 Tudor mutation on TIRR interaction. The 53BP1 mutation W1495A, Y1523A, and Y1500A decreases the binding affinity of TIRR for ~17-fold, ~32-fold, and ~14-fold, and the mutation of Y1502A and Y1523S completely abolished TIRR interaction. These demonstrated the crucial roles of hydrophobic residues locating at the TIRR/53BP1 Tudor-binding interface (Fig. 3c, Table 3). Moreover, TIRR binding was reduced by ~6-fold by D1521A and thoroughly abrogated by D1521R (Fig. 3c, Table 3), suggesting the importance of charge–charge interactions. These results are in agreement with the previous study, in which W1495 and D1521 mutations impaired the interaction between 53BP1 and TIRR in vivo. Located outside of the aromatic cages, 53BP1 Tudor residues L1547 and F1553 in the L1-loop displays interaction with TIRR residue P105 and H106 (Fig. 2c). The binding affinity of TIRR was decreased by 53BP1 residue L1547A, L1547G for ~5-fold ($K_d$ of 4.7 μM) and ~19-fold ($K_d$ of 17.3 μM), respectively (Fig. 3c, Table 3). In contrast, F1553A slightly decreased the TIRR-binding affinity for ~2-fold ($K_d$ of 1.5 μM). These results strongly suggest that the residues in L1-loop of Tudor domain also contribute to TIRR binding, albeit to a less extent. Intriguingly, Y1523A disrupts the 53BP1 Tudor binding to TIRR without impacting binding to H4K20me2 (Supplementary

| Table 1 Data collection and refinement statistics | |
| --- | --- |
| | **53BP1 & TIRR** |
| Data collection | |
| Space group | P65 |
| Cell dimensions | |
| $a, b, c$ (Å) | 167.11, 167.11, 46.51 |
| $\alpha, \beta, \gamma$ (°) | 90, 90, 120 |
| Resolution (Å) | 50.00-1.76 (1.82-1.76) |
| $R_{merge}$ | 0.107 (2.258) |
| $R_{pim}$ | 0.023 (0.645) |
| $I/\sigma I$ | 74.8 (1.9) |
| Completeness (%) | 99.9 (100.0) |
| Redundancy | 19.7 (18.7) |
| Refinement | |
| Resolution (Å) | 40.64-1.76 |
| No. reflections | 73,765 |
| $R_{work}/R_{free}$ | 0.20 / 0.22 |
| No. atoms | |
| Protein | 4092 |
| Water | 118 |
| $B$-factors | 39.2 |
| Protein | 39.4 |
| Water | 35.2 |
| R.m.s. deviations | |
| Bond lengths (Å) | 0.008 |
| Bond angles (°) | 1.203 |

Values in parentheses are for highest-resolution shell. One crystal was used for the data

**Fig. 1** Characterization of the TIRR/53BP1 Tudor complex. **a** Schematic presentation of 53BP1 and TIRR prepared using IBS software[30]. The 53BP1 N-terminal region containing 28 × S/TQ sites, the 53BP1 Tudor domain, UDR domain, and TIRR are either labeled or colored in magenta, orange, and cyan, respectively. The secondary structure of 53BP1 double Tudor and TIRR are presented, with the interaction regions of Tudor and TIRR highlighted at the bottom. **b** Isothermal titration calorimetric analysis of TIRR binding to 53BP1 Tudor or Tudor-UDR. TIRR is titrated by 53BP1 Tudor (left) and 53BP1 Tudor-UDR (right). **c** Sedimentation coefficients for 53BP1 Tudor (magenta) and TIRR (cyan), and the complex formed by 53BP1 Tudor and TIRR in 2:2 ratio (black). The sedimentation coefficient distributions c(s) are plotted in relative absorbance units $A_{280}$ vs. svedbergs (S). **d** The overall structure of TIRR in complex with the 53BP1 Tudor domain. TIRR protomers are shown by cyan and green, respectively. The 53BP1 tandem Tudor domain is colored in magenta. The N-terminal region and L8-loop of TIRRs (in Tudor-bound or unbound forms), along with the L1, L3, L1′ loops of 53BP1 Tudor are indicated

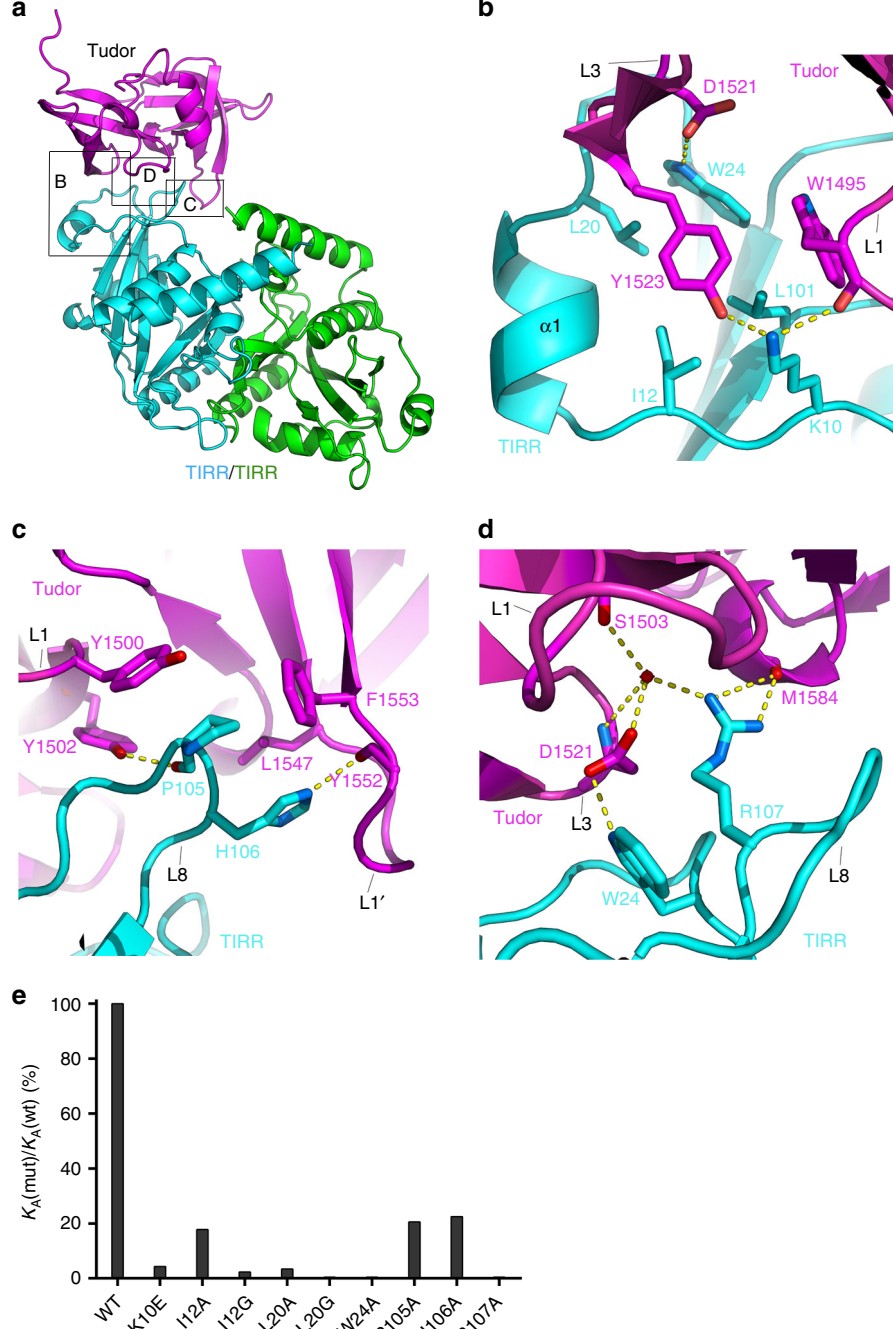

**Fig. 2** Structure of TIRR in the complex of the 53BP1 tandem Tudor domain. **a** The overall view of TIRR/53BP1 Tudor complex. Highlighted are zoom-in regions for close-up view. **b–d** Close view of intermolecular contacts between 53BP1 Tudor and TIRR. Highlighted are 53BP1 Tudor residues interacting with TIRR through the N-terminal region (**b**), the L8-loop (**c**), and both regions (**d**). 53BP1 Tudor and TIRR are colored in magenta and cyan, respectively. **e** Effect of TIRR single-residue mutation on the binding of 53BP1 Tudor assayed by ITC

Fig. 4B). The 53BP1 Y1523A mutant may function normally in recognizing H4K20me2, notwithstanding the loss of TIRR binding.

In conclusion, TIRR binding alters the structure of the 53BP1 Tudor aromatic cage used for H4K20me2 recognition. These findings reveal how TIRR serves as a natural Tudor inhibitor and regulates the H4K20me2-binding function of 53BP1.

**Interaction of TIRR and 53BP1 Tudor regulates 53BP1 activity.** To validate the effect of critical TIRR residues on the 53BP1 binding, we generated four mutants of TIRR including K10A,

W24A, H106A, and R107A. As shown in Fig. 4a, all TIRR mutants abolish the interaction with 53BP1, suggesting that these residues of TIRR are responsible for its interaction with 53BP1. In addition, similar to 53BP1 D1521R, we observed that 53BP1 mutations Y1502A and Y1523A severely impair the interaction of 53BP1 and TIRR (Fig. 4b), indicating that these residues within Tudor domain are critical for the intact 53BP1 protein function. Since previous studies demonstrated that TIRR is critical for 53BP1 stability[19], we next investigated how these mutants affect the protein level of 53BP1 in TIRR-knockout cells, given that none of the TIRR point mutants interacts with 53BP1. Indeed, loss of TIRR led to a decrease of 53BP1 protein levels (Fig. 4c).

| Table 2 Effects of TIRR mutation on binding of 53BP1 Tudor | |
| --- | --- |
| **TIRR** | **Binding affinity ($K_d$, μM)** |
| WT | 0.9 |
| K10E | 20.4 |
| I12A | 4.9 |
| I12G | 38.3 |
| L20A | 25.5 |
| L20G | ND |
| W24A | ND |
| P105A | 4.3 |
| H106A | 3.9 |
| R107A | ND |
| *ND* not detected | |

Moreover, expression of gRNA-resistant wt TIRR, but not the TIRR mutants, restored 53BP1 protein levels in TIRR-knockout cells (Fig. 4c), indicating that the 53BP1–TIRR interaction is required for 53BP1 expression. These results are consistent with our ITC results (Fig. 3c) and strongly suggest that residues within the binding interface of TIRR and 53BP1 Tudor are critical for their in vitro and in vivo interactions.

TIRR has been reported to participate in DNA repair pathway[18,19]. We observed that Replication protein A2 (RPA2) foci formation was reduced in TIRR-knockout cells, and the TIRR-deficient cells were sensitive to IR. Expression of gRNA-resistant wt TIRR re-establishes the RPA2 foci (Fig. 4d) and compromises the sensitivity of TIRR-knockout cells to IR (Fig. 4e). However, reconstitution with TIRR mutants with reduced 53BP1 binding affinity fail to rescue RPA2 foci formation in TIRR-knockout cells (Fig. 4d). Consistently, TIRR-deficient cells expressing TIRR mutants are sensitive to IR (Fig. 4e). These results suggested that TIRR contributes to DNA repair at least in part via its interaction with 53BP1. A crosstalk of TIRR and 53BP1 has been implicated by early studies, which demonstrated the inhibitory effect of TIRR on 53BP1 recruitment at DNA damage sites after irradiation[18,19]. We observed 53BP1 foci formation in cells overexpressing the TIRR mutants after IR. It is in sharply contrasted to the previous observation, in which 53BP1 foci formation is markedly inhibited in cells overexpressing the wt TIRR (Fig. 4f, g). Our results suggested that TIRR interaction with 53BP1 prevents 53BP1 localization to sites of DNA damage.

We next examined 53BP1 Y1523A, a separation-of-function mutant exclusively influencing TIRR–Tudor interaction, for its effect on IR-induced 53BP1 foci formation and RIF1(Rap1 interacting factor 1) accumulations at DSB sites. Our results showed in MCF10A-derived 53BP1-knockout cells, the 53BP1 D1521R mutation, which disrupts this activity of the Tudor domain, failed to form IR-induced foci, whereas 53BP1 Y1523A formed foci that colocalized with γ-H2AX (Supplementary Fig. 5A). It is well known that RIF1 is a downstream effector of 53BP1 during DSB repair, and RIF1 accumulations at DSB sites is dependent on 53BP1 Tudor domain. As shown in Supplementary Fig. 5B, the RIF1 foci is abolished in 53BP1-knockout cells, while RIF1 foci is re-established in the cells overexpressing 53BP1 wt or Y1523A mutant, but not the D1521R mutant. Furthermore, foci formation of 53BP1 Y1523A is still observed in cells overexpressing TIRR, however, TIRR expression almost completely abolishes foci formation of 53BP1 wt (Supplementary Fig. 5C).

**TIRR residues distinct from Nutd16 dictate Tudor recognition.** TIRR shares high sequence homology and overall structural

similarity with Nudt16[23], a member of the Nucleoside diphosphate-linked moiety X (NUDIX) hydrolase family[20] (Supplementary Fig. 6). Our ITC analysis revealed that Nudt16 does not interact with 53BP1 Tudor domain (Fig. 5c), indicating that some particular Nudt16 residues prevent Nudt16 recognizing 53BP1 Tudor. Structure comparison reveals that TIRR residues responsible for 53BP1 Tudor binding are largely conserved in Nudt16 (The counterpart of residues R5, L7, L15, W19, and V100 in Nut16 are K10, I12, L20, W24, and L101 in TIRR, respectively) (Fig. 5a, b). The only exception is a TIRR histidine (residue H106) which is exclusively presented in L8-loop of TIRR (Fig. 5b). Instead, Nudt16 lacking histidine contains a shorter L8-loop, which likely perturbs the Nudt16 binding of the second Tudor domain. To investigate the effect of this histidine residue on TIRR–Tudor interaction, we generated mutants of TIRR and Nudt16, and measured their binding to 53BP1 Tudor by ITC analyses. On the one hand, deletion of H106 in TIRR (H106Δ) dramatically decreased the binding affinity of 53BP1 Tudor by a factor of ~20 ($K_d$ of 17.8 μM) (Fig. 5c, Table 4). On the other hand, a Nudt16 mutant with histidine insertion (Nudt16 ^H105, insertion of residue is designate as ^ hereafter) yields a binding affinity of 7.1 μM. It is a remarkable binding increase given that the binding of Nudt16-Tudor is undetectable (Fig. 5c). Furthermore, substitution of residues R5, L7, V100 of Nudt16 ^H105 with TIRR residues K10, I12, and L101 yields a $K_d$ of 1.0 μM, and largely compromises the binding losses of Nudt16-Tudor (Fig. 5c). We concluded that the impaired binding of Nudt16 for 53BP1 Tudor is attributed to two factors: the Nudt16-specific residues R5, L7, and V100, and the absence of TIRR histidine in L8-loop. Substitution of Nudt16 with TIRR residues H106, K10, I12, and L101 are both necessary and sufficient to confer Nudt16, an ability of 53BP1 Tudor binding.

Structure-based sequence alignments showed that TIRR residue H106 is highly conserved in mammals and birds which have bona fide TIRR proteins (Supplementary Fig. 7A). However, in a number of species, the histidine is replaced by a glutamine (Supplementary Fig. 7B). This suggested that (1) TIRR-containing Q106 may function normally as TIRR-containing H106 does, and (2) the presence of TIRR H106, rather than the type of H106, is more crucial for TIRR function. To validate these assumptions, we replaced H106 with the polar or hydrophobic residues and investigated the effects of these mutations on 53BP1 Tudor binding. In comparison with the dramatic binding affinity loss for TIRR ΔH106 ($K_d = 17.8$ μM, ~20-fold of reduction), TIRR mutants H106A, H106Q, and H106K displayed modest binding affinity decreases (~4.4, ~5.8, ~7.7-fold of reduction), whereas mutants H106R and H106Y only display mild decreases (~2.1 and ~1.5-fold of reduction) (Fig. 5d, Table 4). These results also suggest that mutations of TIRR H106 could modulate the binding of TIRR–53BP1 Tudor, likely by fine-tuning their contact interface. We postulated that TIRR mutant H106W could improve 53BP1 Tudor binding because a tryptophan residue retains the hydrogen bond formation ability and provides more hydrophobic contact than histidine does (Supplementary Fig. 8A, B). Indeed, ITC results demonstrated that TIRR H106W displays ~2-fold higher bind affinity ($K_d$ of 0.5 μM) than TIRR wt (Fig. 5d, Table 4), and addition of an extra tryptophan to Nudt16 (^W105) yields a $K_d$ of 3.6 μM for 53BP1 Tudor binding, representing a ~2-fold increase, compared to Nudt16 ^H105 (Supplementary Fig. 8C).

Collectively, these results suggest that the binding of TIRR–Tudor can be modulated and TIRR can serve as an ideal candidate for Tudor inhibitors screening. The identification of TIRR-mutant H106W, which shows the strongest 53BP1 Tudor binding to date, provides a first proof-of-concept demonstration

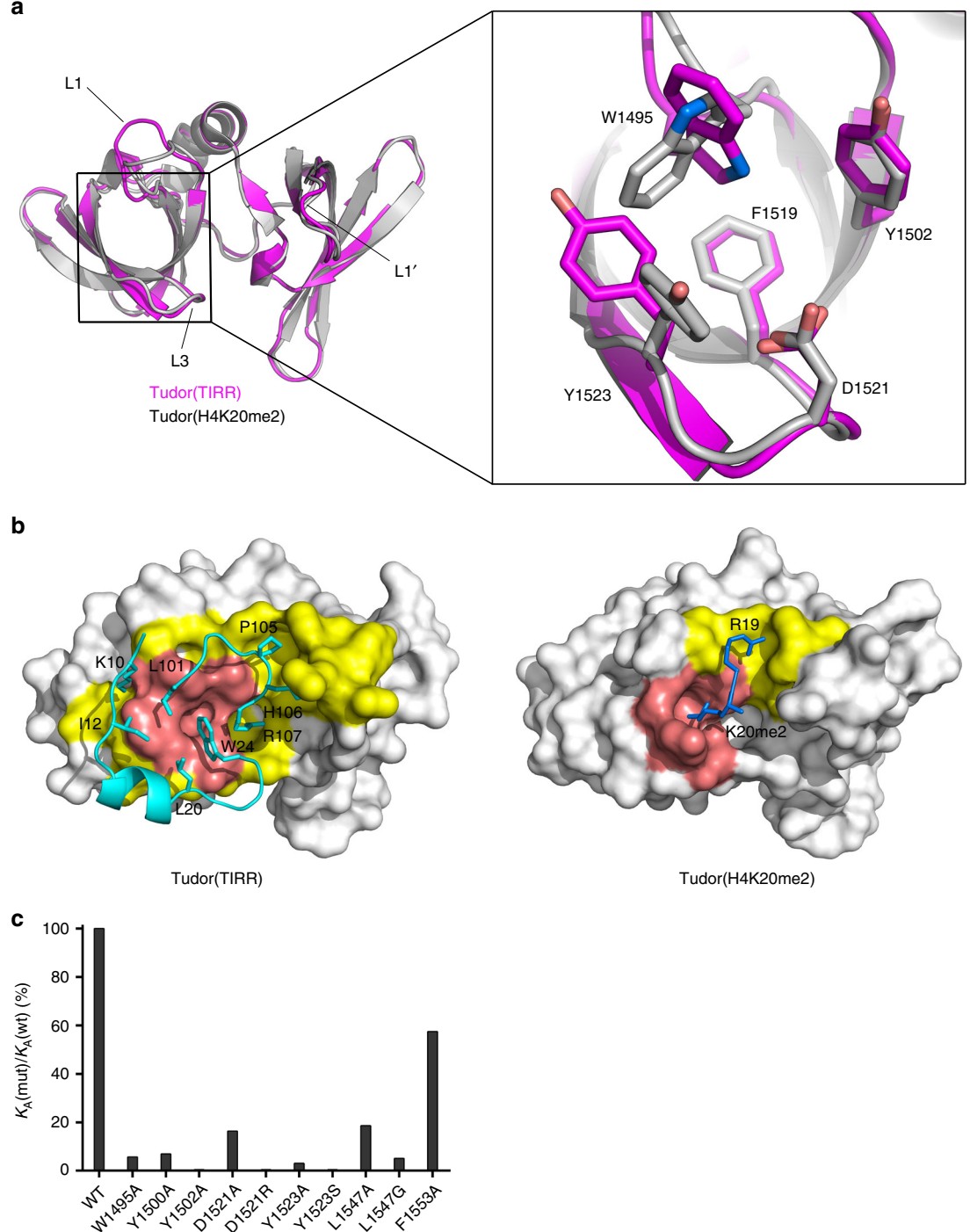

**Fig. 3** Structure of the 53BP1 tandem Tudor domain in the complex of TIRR. **a** Structural comparison of 53BP1 Tudors in complex with TIRR (magenta) or H4K20me2 (gray, PDB 2IG0) (left) and close view of methylation-reading aromatic cages in superimposition (right). Highlighted are 53BP1 residues W1495, Y1523, D1521, Y1502, and F1519 forming the aromatic cage. **b** Surface representation of 53BP1 Tudor structures in TIRR-bound form (left) and H4K20me2-bound form (right). Residues of TIRR and histone H4K20me2 involved in Tudor interaction are displayed and colored in cyan and blue, respectively. Highlighted are 53BP1 residues forming the aromatic cage (salmon), and other 53BP1 residues involved in TIRR and H4K20me2 binding (yellow). **c** Effect of 53BP1 Tudor mutations on TIRR binding analyzed by ITC

of how to create the novel 53BP1 inhibitors by structure-based rational design.

## Discussion

TIRR modulates the chromatin recruitment of 53BP1, a key factor that functions in the choice of DSB repair pathway. In this study, we provide the structural basis of 53BP1 recognition by TIRR and elucidate the mechanism of TIRR acting as the bona-fide cellular Tudor inhibitor. TIRR uses its N-terminal region coupled with L8-loop to block the methylation reader binding surface in Tudor, thus abolishing the access of 53BP1 to nucleosome bearing H4K20me2. Our data are completely consistent with results of the previous study showing that TIRR

| Table 3 Effects of 53BP1 Tudor mutation on TIRR binding | |
| --- | --- |
| **53BP1-Tudor** | **Binding affinity ($K_d$, µM)** |
| WT | 0.9 |
| W1495A | 15.4 |
| Y1500A | 12.7 |
| Y1502A | ND[a] |
| D1521A | 5.4 |
| D1521R | ND |
| Y1523A | 28.5 |
| Y1523S | ND |
| L1547A | 4.7 |
| L1547G | 17.3 |
| F1553A | 1.5 |
| [a]ND means not detected | |

regulates the function of 53BP1 by recognizing the histone methyl-lysine binding site in 53BP1 Tudor. Nevertheless, TIRR mutants with 53BP1 binding defects fail to regulate 53BP1 activity in vivo, underscoring the important roles of TIRR in DSB repair control via 53BP1.

The previous study mapped the binding interface of TIRR and 53BP1 Tudor by NMR and revealed a preferential disappearance of $^1H$-$^{13}C$ HMQC spectra signals assigned to K10 and K151, suggesting that they are in the proximity of the binding interface. However, in comparison with TIRR residue K10, residue K151 appears less important for 53BP1 Tudor function because a single K151E mutation exhibits no effect on 53BP1 interaction in vivo[18]. Indeed, our structure demonstrated that TIRR residue K10 directly interacts with 53BP1 Tudor, whereas TIRR residue K151 is located in close proximity to the binding interface of TIRR/53BP1 Tudor (Supplementary Fig. 6B). Binding of TIRR K151 is likely dynamic and regulated by the accessibility of 53BP1 Tudor. The dynamics of TIRR and 53BP1 Tudor could be attributed from the intrinsic flexibility of the local structure. In agreement with this assumption, poor electron densities are observed for residues P105 and H106 in human TIRR structure (PDB 3KVH), and W1495 and Y1523 in 53BP1 Tudor (PDB 2G3R) structure[12].

The observed intrinsic flexibility of 53BP1 Tudor implies the nature of conformational changes. While TIRR binding induces a conformational change in 53BP1 Tudor, the H4K20me2-bound 53BP1 Tudor closely resembles the 53BP1 Tudor in free form, suggesting that binding of H4K20me2 causes undetectable conformational changes. Compared to H4K20me2-bound Tudor, binding of 53BP1 Tudor to TIRR results in a substantially increased contact area and yields a higher binding affinity (Supplementary Fig. 4). These results strongly suggest that the conformational changes greatly contribute to the complex stability. We postulated that the robust binding of 53BP1 Tudor to TIRR may block the ubiquitin-targeted degradation, which ensues after 53BP1 released from the chromatin epitopes or other binding factors. TIRR therefore can regulate 53BP1 biogenesis by altering the stoichiometry and dynamics of 53BP1 interacting with DSBs[18,19].

It is reported that TIRR (alias Nudt16L1) originated from the ancient Nudt16 gene due to mutation of the DNA[23]. TIRR lacks an intact NUDIX domain, which is required for Nudt16 hydrolysis activity, representing the largest sequence divergence between these two paralogs (Supplementary Fig. 6). The specificity of 53BP1 Tudor recognition by TIRR is unlikely conferred by NUDIX domain because this domain is located far from the binding interface of TIRR/53BP1 Tudor, as shown by the complex structure (Supplementary Fig. 6). Conversely, a previously uncharacterized TIRR residue H106 acts as a bonafide factor that confers binding of 53BP1 Tudor to TIRR. The appearance of this

histidine, which is not observed in Nudt16, displays a remarkable conservation in the TIRR of the mammals and the birds (Supplementary Fig. 7A). Our findings reveal an evolutionary conserved mechanism for 53BP1 Tudor recognition by TIRR.

The previous study revealed a clinical relevance of TIRR in cancer therapy because TIRR controls the cellular function of 53BP1 by regulating its accumulation at damaged chromatin[18,19]. Increasing TIRR protein level inactivates 53BP1 and causes BRCA1-mutant tumors resistant to PARPi[18,19]. Moreover, inhibition of histone methyl-lysine readers containing Tudor domain is of great importance because many of the readers are associated with the pathogenesis of the various disease. Our studies reveal TIRR residues that influence the binding of 53BP1 Tudor and generate TIRR mutants (H106W) with improved Tudor interaction. These results, as a proof-of-concept, demonstrate the feasibility of creating TIRR mutant that can inhibit the methyl-lysine binding function of 53BP1 and other readers. Importantly, TIRR mutants with altered 53BP1 Tudor binding may sensitize the BRCA1-mutant tumor cells to PARPi, indicating a therapeutic potential of TIRR in cancer treatment.

## Methods

**Protein expression and purification**. Coding sequences for *M. musculus* TIRR (6–211), *H. sapiens* TIRR(6–211), *H. sapiens* Nudt16(1–195), *H. sapiens* 53BP1 Tudor domain (1484–1603), and Tudor-UDR domain (1484–1631) were amplified from cDNA and inserted into pET28 expression vectors containing an N-terminal 6 × his tag and Tobacco Etch Virus (TEV) protease cleavage site(leaving an N-terminal overhang of the residues Gly-His-Met). A list of PCR primers used in this study is provided in Supplementary Tables 1-3. Constructs were transformed and grown in *Escherichia coli* BL21-Codon PLUS(DE3)-RIPL cells(Stratagene) to an $A600$ of ~0.8 and induced with 0.5 mM isopropyl β-ᴅ-thiogalactoside (IPTG) in Luria-Bertani (LB) broth at 18 °C for 16 h. Subsequently, the cells were harvested by centrifugation and stored at −20 °C. The 53BP1 Tudor and Tudor UDR were purified as reported previously[12] with some modifications. Briefly, the cells were suspended and lysed in buffer A (20 mM HEPES, pH 7.5, 500 mM NaCl, 5% glycerol, 5 mM imidazole, and 5 mM BME). Lysate was clarified by centrifugation and the supernatant was passed through a Ni-NTA column(Qiagen) equilibrated with buffer A, and the protein was eluted in buffer B (20 mM HEPES, pH 7.5, 500 mM NaCl, 5% glycerol, 250 mM imidazole, and 5 mM BME). The His-tag was removed by overnight digestion with TEV protease at room temperature. A nickel agarose column was used to remove the TEV protease and digested His-tag, and the flow-through was concentrated and loaded onto a 24 mL size-exclusion column Superdex200 (GE Healthcare) equilibrated with buffer C (20 mM HEPES, pH 7.5, 150 mM NaCl, and 1 mM DTT) for final purification. The purity of the 53BP1-Tudor was over 95% analyzed by SDS-PAGE. Nudt16 and TIRR were first purified by immobilized metal affinity chromatography (IMAC) using Ni-NTA agarose resin (Qiagen) equilibrated with buffer D (50 mM Tris-HCl, pH 8.0, 500 mM NaCl, 10% glycerol, 10 mM imidazole, and 5 mM BME). After several rounds of washing, target proteins were eluted in buffer E (20 mM Tris-HCl, pH 8.0, 500 mM NaCl, 10% glycerol, 500 mM imidazole, and 5 mM BME) and processed by overnight incubation with TEV protease at room temperature. A nickel agarose column was used to remove the tagged TEV protease and unprocessed protein, and the flow-through was further purified by application to a 5 ml Hitrap Heparin column(GE Healthcare) and elution over a NaCl gradient. Fractions containing TIRR or Nudt16 were pooled, concentrated, and loaded onto a 24 mL Superdex200 size-exclusion column (GE Healthcare) in buffer F (20 mM Tris-HCl, pH 7.5, 300 mM NaCl, 10% glycerol, and 1 mM DTT). The purified target proteins were stored at −80 °C for further use. Mutants were prepared using the QuikChange Site-Directed Mutagenesis Kit (Stratagene), and the sequences were verified by sequencing. Mutants of 53BP1 Tudor, TIRR and Nudt16 were expressed and purified, as described for the wt protein.

**Crystallization and structure determination**. The human and mouse TIRR were mixed with 53BP1 Tudor domain in different ratio and the protein complexes were used for crystallization attempts at 16 °C with the hanging-drop vapor-diffusion method. The complex crystals grew in drops containing 1 µl protein (18.5 mg/ml) and 1 µl reservoir solution, against 200 µl reservoir solutions containing 0.1 M Bis–Tris propane, pH 9.0, and 18–22% PEG MME550 (all reagents from Sigma). The crystals were flash cooled in liquid nitrogen with a cryoprotectant, prepared from the reservoir solution supplemented with 20% glycerol. X-ray diffraction data were collected on the beamline BL17U with a Quantum 315r CCD detector (ADSC) at Shanghai Synchrotron Radiation Facility (SSRF) and HKL2000 package[24] was used for data processing, integration, and scaling. The correct solution is achieved via the molecular replacement method in PHASER[25], using the crystal structure of 53BP1 Tudor (PDB 2G3R)[12] and mouse TIRR (PDB 4ZG0)[21] as the

initial search model. The COOT[26] and PHENIX[27] were used for manual model building and refinement, whereas PROCHECK[28] was used for model geometry verification. The final structure of this complex was refined to 1.76 Å resolution, against $R_{work}/R_{free} = 0.20/0.22$. The statistics for data collection and structural refinement are summarized in Table 1, and figures were prepared with PyMOL (http://www.pymol.org/).

**Isothermal titration calorimetry.** All ITC measurements were recorded with an ITC200 titration calorimeter (MicroCal). Purified recombinant proteins were dialyzed overnight against ITC buffer (20 mM Tris-HCl, pH 7.5, 150 mM NaCl) and then diluted with the same buffer to achieve the desired concentrations: 22–44 µM TIRR, Nudt16, and 200–400 µM 53BP1 Tudor domain. Titrations for all reactions were done at 25 °C, including an initial injection of 0.4 µl (which was

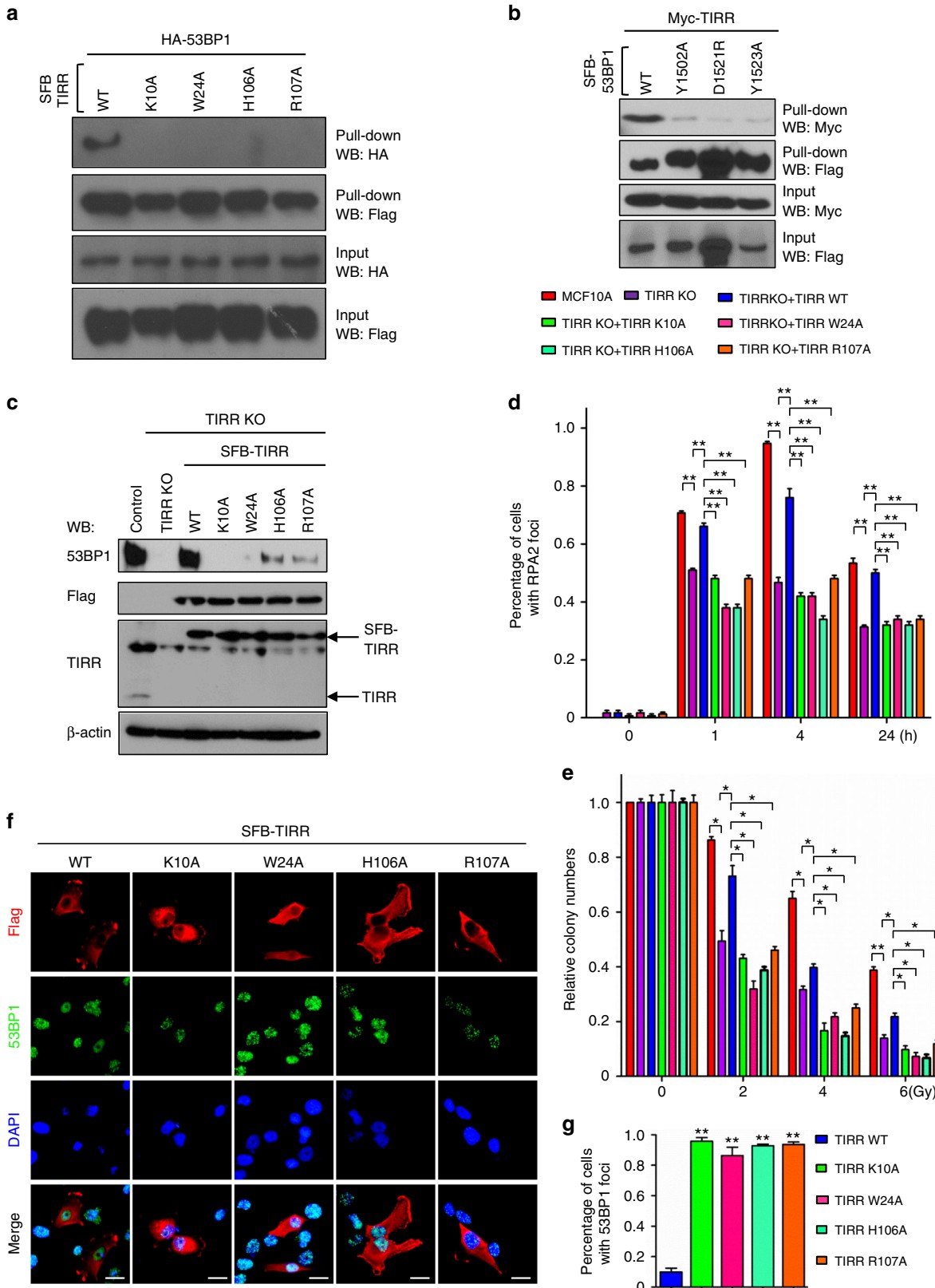

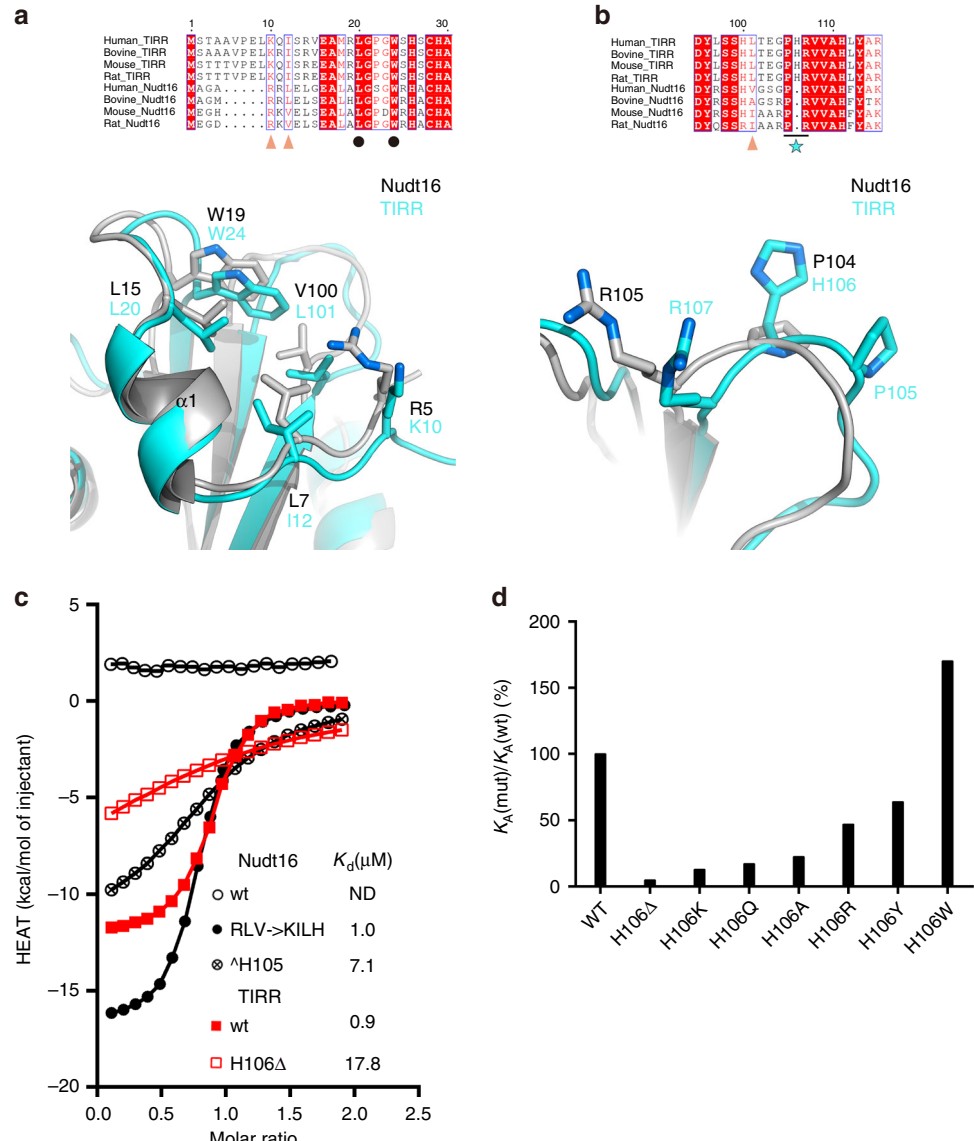

**Fig. 5** TIRR residues confer a specific recognition for 53BP1 Tudor. **a**, **b** Sequence and structural divergence of TIRR and Nudt16 in the N-terminal region (**a**) and the L8-loop region (**b**). Top panel: Multiple sequence alignment of TIRR and Nudt16 from human, bovine, mouse, and rat. Lower panel: Superimposed structures of TIRR (cyan) and Nudt16 (gray, PDB 3COU). TIRR residues participating in 53BP1 Tudor binding are colored in orange. Triangles and circles represent residues that are either conserved or non-conserved between TIRR and Nudt16. The blue star represents the histidine that is exclusively present in TIRR. **c** Isothermal titration calorimetric analysis of 53BP1 Tudor binding to wild-type or mutated TIRR (Nudt16). Representative binding results for Nudt16 and TIRR are highlighted in black and red, respectively. wt (red filled rectangles) and TIRR H106Δ(red open rectangles). The binding affinity is shown accordingly. **d** Effects of TIRR H106 mutation on bindings of 53BP1 Tudor

**Fig. 4** TIRR regulates 53BP1 activity in vivo. **a** TIRR mutants disrupt the interaction with 53BP1. A total of 293T cells were transfected with plasmids encoding SFB-tagged wild-type TIRR or the TIRR mutants together with the plasmid encoding HA-tagged wild-type 53BP1. Immunoprecipitation reactions were conducted using Streptavidin-protein beads and then subjected to western blotting using the indicated antibodies. The full blot is shown in Supplementary Fig. 9. **b** 53BP1 mutants impair the binding affinity to TIRR. A total of 293T cells were transfected with plasmids encoding SFB-tagged wild-type 53BP1 or the 53BP1 mutants together with the plasmid encoding Myc-tagged wild-type TIRR. Immunoprecipitation reactions were conducted using S beads and then subjected to western blotting using the indicated antibodies. The full blot is shown in Supplementary Fig. 9. **c** TIRR mutants affect the stability of 53BP1. TIRR-knockout cells were reconstituted with wt TIRR or indicated mutants of TIRR. The indicated cell lines were collected and were immunoblotted with indicated antibodies. The full blot is shown in Supplementary Fig. 9. **d** TIRR mutants impair DNA repair function. TIRR-knockout cells were reconstituted with wt TIRR or indicated mutants of TIRR. The indicated cell lines were treated with IR and were processed for RPA immunofluorescence at indicated time point post IR. RPA foci were quantified (at least 400 cells were counted for each experiment). Data are represented as the mean ± s.d. ($n = 3$). $*p < 0.05$; $**p < 0.01$. **e** TIRR mutants sensitize the cells to ionizing radiation treatment. TIRR knock-out cells were reconstituted with wt TIRR or indicated mutants of TIRR. The indicated cell lines were treated with various doses of IR. Cell survival following irradiation was measured by clonogenic assay according to the "Experimental Procedures." $*p < 0.05$; $**p < 0.01$. **f**, **g** TIRR mutants abolish its inhibitory effect on 53BP1 recruitment at DNA-damage sites after IR. **f** MDA231 breast cancer cells were transfected with plasmids encoding SFB-tagged wt or mutants of TIRR. After 24 h of transfection, cells were subjected to IR. Cells were fixed 1 h later and the immunofluorescence was performed with indicated antibodies. Scale bar represents 200 μm. **g** Quantitative analysis of 53BP1 foci at DSBs in cells overexpressing SFB-tagged wt or mutants of TIRR. $**p < 0.01$

**Table 4 Effects of TIRR H106 mutation on binding of 53BP1 Tudor**

| TIRR | Binding affinity ($K_d$, μM) |
|---|---|
| WT | 0.9 |
| H106Δ | 17.8 |
| H106K | 6.8 |
| H106Q | 5.1 |
| H106A | 3.9 |
| H106R | 1.9 |
| H106Y | 1.4 |
| H106W | 0.5 |

omitted from data analysis), followed by 19 sequential 2.0 μl injections of the 53BP1 Tudor domain (wt or mutant) into 200 μl of the TIRR (wt or mutant) and Nudt16 (wt or mutant), spaced at intervals of 120 s. In all cases, three independent experiments were performed. The raw ITC data were processed with Origin 7.0 software (Microcal) and the curves were fitted to a single-site binding model. Notably, the protein mutants were well behaved and no aggregation of mutant proteins was observed during the experiment.

**Analytical ultracentrifugation**. All analytical ultracentrifugation experiments, including the sedimentation velocity and sedimentation equilibrium analysis, were performed on a Beckman Coulter Proteome Lab XL-I analytical ultracentrifuge with An-60Ti rotor (Beckman). The samples for analytical ultracentrifugation were dialyzed against 20 mM Tris-HCl, pH 7.5, 150 mM NaCl, and initial absorbances at 280 nm were adjusted to approximate 0.8. The samples were equilibrated for 6 h at 20 °C under a vacuum in a centrifuge before sedimentation. The absorbance at 280 nm was measured in continuous-scan mode during sedimentation in aluminum double-sector cells at 20 °C, $290,000 \times g$ (sedimentation velocity) or at 16 °C, $26,080 \times g$ (sedimentation equilibrium). The Sedfit were used for data analysis (http://www.analyticalultracentrifugation.com/).

**Antibodies**. Anti-TIRR antibody was described previously[19]. The monoclonal purchased were shown as below: anti-FLAG M2 (Sigma F3165), anti-β-actin (Sigma A5441), anti-HA (Invitrogen 26183), and anti-Myc (9E10) (Santa Cruz sc-40). All antibodies used in this study are depicted in Supplementary Table 4.

**Cell culture and plasmids**. HEK293T, MCF10A, and MDA-MB-231 cells were purchased from American Type Culture Collection (ATCC), which were cultured under conditions specified by the manufacturer. The cell lines have been tested for mycoplasma contamination. TIRR cDNA or 53BP1 cDNA was subcloned into pDONR201 as entry vector, then they were subsequently transferred to gateway-compatible destination vectors for the expression of triple-epitope tag SFB, HA, or Myc epitope-tagged fusion proteins. All deletion mutants were constructed by site-directed mutagenesis and verified by sequencing.

**Immunofluorescence staining**. Cells were cultured on coverslips and treated with 20 Gy irradiation, followed by recovery for 4 h or as indicated time. Cells were then washed with PBS and incubated in 3% paraformaldehyde solution for 10 min at room temperature. Cells were then washed with PBS and permeabilized in 0.5% Triton X-100-containing solution for 5 min at room temperature. Samples were incubated with primary antibodies diluted in 5% goat serum at room temperature for 2 h. After washing with PBS, the samples were incubated with secondary antibody for 1 h at room temperature. Samples were then counterstained with DAPI and mounted onto glass slides with anti-fade solution. Samples were visualized with a Leica fluorescence microscope.

**Co-immunoprecipitation and western blotting**. Cells were lysed in NETN buffer (100 mM NaCl, 1 mM EDTA, 20 mM Tris-HCl at pH 8.0, and 0.5% NP-40) with protease inhibitors. Cell lysates were centrifuged at 14,000 rpm for 15 mins at 4 °C. For precipitation of SFB-tagged proteins, cell lysates were incubated with strep-tavidin beads (GE Healthcare) for 3 h at 4 °C. After washing with NTEN buffer, the immunocomplexes were eluted by boiling in 2× laemmli buffer. Samples were resolved by SDS-PAGE, transferred to PVDF membranes, and immunoblotted with indicated antibodies.

**Clonogenic survival assays**. IR sensitivity assay was carried out as described previously[29]. Briefly, cells were seeded onto 60-mm dish in triplicates and treated with IR with indicated doses. Cells were left for 14 days to allow colonies to form. Colonies were fixed and stained with Coomassie blue and then counted using a GelDoc with Quantity One software (BIORAD). The results were the averages of the data obtained from three independent experiments.

**Data availability**. Data supporting the findings of this manuscript are available from the corresponding authors upon reasonable request. The coordinates and structure factors for TIRR/53BP1 Tudor complex were deposited into the PDB under accession code 5Z78.

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

## Acknowledgements

We thank SSRF beamline scientists for technical support on data collection, Yuanyuan Chen, Zhenwei Yang (Institute of Biophysics, Chinese Academy of Sciences) for technical help with ITC analysis, Xiaoxia Yu for assistance on AUC analysis, Linchang Dai and Jujun Zhou for advice on structure determination. Without their support, this work would not be possible. The study was supported by grants from the Chinese Ministry of Science and Technology (2015CB856200 to Z.Z.), grants from the Natural Science Foundation of China (31521002, 31671344 to Z.Z., 31600623 to S.S.), the Strategic Priority Research Program (XDB08010104 to Z.Z.) and an Ovarian Cancer Research Fund Alliance Grant (373376 to Z.G.), and an NCI grant (CA192052 to Z.G.). Z.Z. is supported as a CAS-Novo Nordisk Great Wall Professor.

## Author contributions

Y.D. and Z.Z. conceived and initiated the study. Y.D. and A.Z. performed all the experiments under the supervision of Z.G. and Z.Z. S.S. determined the structure. Y.D., Z.G., and Z.Z. analyzed the data, prepared the figures, and wrote the manuscript. All authors contributed to experimental design and commented on the manuscript.

## Additional information

**Competing interests:** The authors declare no competing interests.

