## [Peer Review File · Nature Communications]

Reviewers' comments:

Reviewer #1 (Remarks to the Author):

TIRR is a newly identified regulator of 53BP1, which is a key DNA repair enzyme in the NHEJ pathway. In this paper, the authors determine the high resolution crystal structure of the TIRR-53BP1 Tudor domain complex, revealing an interaction that blocks the methyl-lysine binding pocket of the Tudor domain. Extensive mutagenesis reveals the relative importance of residues at the contact surface, and cellular experiments using these mutants reveals that these surfaces are required for TIRR-53BP1 interactions in human cells, and for the ability of TIRR to regulate DNA damage signaling and repair. The structure and further mutagenesis is used to show why the TIRR homolog Nudt16 does not bind 53BP1. Overall, the quality of the work is high. While the structures of the individual proteins were previously known, and other work had suggested that the interaction would block methyl-lysine binding, I believe that the complex structure and the associated validation work merits serious consideration for publication in Nat. Comm.

Comments/suggestions:

- The cellular work relies on the fact that mutations that destabilize the TIRR-53BP1 interface in vitro also disrupt function in cells. While this is solid data, it does not necessarily rule out other functions for this surface of the TIRR protein, other than 53BP1 binding. Can the authors design second site suppressor mutations that could be used to more specifically test the hypothesis that it is the disruption of the TIRR-Tudor interaction that is responsible for the loss of DNA repair activity? For example, is D1521R rescued by a charge reversal mutation at TIRR R107?
- The unbound form of the 53BP1 tudor domain closely resembles the H4K20me2 form (compare 2ig0 with 2g3r). This may suggest that TIRR binding induces a conformational change in 53BP1 while H4K20me2 seems to bind without need of additional conformational change. This should be discussed.

Minor comments

- Fig. S3 – it looks like density is only displayed for specific sidechains. It might be better to show full electron density for a key region of the interface. While the structure is well refined at high resolution, 2Fo-Fc maps using a bias reduced procedure might be helpful to further demonstrate the accuracy of the map.
- It might be better to include the structure overview in Fig. 2, to help orient the reader to understand where in the overall structure the detailed interactions occur.
- Structure overview (Fig. 1D) – it would be good to indicate where the binding interface for 53BP1 is located in the green, unbound TIRR protomer. This is important to make the case that two molecules of 53BP1 could in theory bind to a TIRR dimer.
- Table 1 – Rmerge is 0 in the highest resolution bin? This must be a typo.
- Some English editing would help with readability of the manuscript.

Reviewer #2 (Remarks to the Author):

In this manuscript, the authors present the crystal structure of TIRR (NUDT16L1) in complex with the Tudor domain of 53BP1. TIRR was recently discovered as a novel 53BP1-binding protein that regulates its activity in DNA repair.

Overall, the structure illuminates the mechanism by which TIRR inhibits 53BP1 and will be of interest to many in the field interested in 53BP1-dependent DNA repair. However, I have a small number of comments/questions that I hope can be answered before publication.

1- While the authors characterize mutations in TIRR that impact its binding to 53BP1, a very useful set of mutations would be mutations, on 53BP1, that selectively disrupt binding to TIRR without

impacting binding to H4K20me2. These mutants will be key to understand the role of TIRR in the regulation of 53BP1. From their analyses, Y1523A or Y1523S might be the mutations that most closely correspond to separation-of-function mutations. What is the impact of these mutations on DNA repair? Are they leading to "hyperactive" 53BP1?

2- The clear observation that the TIRR KO leads to destabilization of 53BP1 suggests that TIRR might function in the biogenesis of 53BP1. Is there any insight from the structure that could explain the destabilization of 53BP1 in the absence of TIRR?

Minor comment

- Ref 11 (Huyen et al.) did not show that 53BP1 bound to H4K20me2. It was rather Botuyan et al. (ref 13).
- I urge the authors to add page numbers.

Referees' comments:

Reviewer #1 (Remarks to the Author):

TIRR is a newly identified regulator of 53BP1, which is a key DNA repair enzyme in the NHEJ pathway. In this papers, the authors determine the high resolution crystal structure of the TIRR-53BP1 Tudor domain complex, revealing an interaction that blocks the methyl-lysine binding pocket of the Tudor domain. Extensive mutagenesis reveals the relative importance of residues at the contact surface, and cellular experiments using these mutants reveals that these surfaces are required for TIRR-53BP1 interactions in human cells, and for the ability of TIRR to regulate DNA damage signaling and repair. The structure and further mutagenesis is used to show why the TIRR homolog Nudt16 does not bind 53BP1. Overall, the quality of the work is high. While the structures of the individual proteins were previously known, and other work had suggested that the interaction would block methyl-lysine binding, I believe that the complex structure and the associated validation work merits serious consideration for publication in Nat. Comm.

Comments/suggestions:

- The cellular work relies on the fact that mutations that destabilize the TIRR-53BP1 interface in vitro also disrupt function in cells. While this is solid data, it does not necessarily rule out other functions for this surface of the TIRR protein, other than 53BP1 binding. Can the authors design second site suppressor mutations that could be used to more specifically test the hypothesis that it is the disruption of the TIRR-Tudor interaction that is responsible for the loss of DNA repair activity? For example, is D1521R rescued by a charge reversal mutation at TIRR R107?

We thank the reviewers for the suggestion. We understand that to find the suppressor mutations that could rescue the disrupted TIRR-Tudor binding will strengthen the manuscript. As suggested by the reviewer, we generated the charge reversal mutations of TIRR R107 and performed ITC analyses to investigate whether these mutants could rescue the TIRR binding disrupted by Tudor D1521R. However, the ITC results showed neither TIRR R107D nor TIRR

R107E binds to Tudor D1521R (See Figures attached below). These suggest that it is challenging to create the gain-of-function mutations that could rescue the disrupted binding of TIRR-Tudor. Ideally, a phage display system can be used for screening the suppressor mutations. This work may be beyond the scope of the current study and will be more appropriate for future exploration.

- The unbound form of the 53BP1 tudor domain closely resembles the H4K20me2 form (compare 2ig0 with 2g3r). This may suggest that TIRR binding induces a conformational change in 53BP1 while H4K20me2 seems to bind without need of additional conformational change. This should be discussed.

The reviewer is correct. As requested, we have addressed this issue in the text (on page 13, 3rd paragraph), as follows “The observed intrinsic flexibility of 53BP1 Tudor implies the nature of conformational changes. While TIRR binding induces a conformational change in 53BP1 Tudor, the H4K20me2-bound 53BP1 Tudor closely resembles the 53BP1 Tudor in free form, suggesting binding of H4K20me2 causes undetectable conformational changes. Compared to H4K20me2-bound Tudor, binding of 53BP1 Tudor to TIRR results in a substantially increased contact area and yield a higher binding affinity (Supplementary Fig. 4). These results strongly suggest that the conformational changes greatly contribute to the complex stability.”

Minor comments

- Fig. S3 – it looks like density is only displayed for specific sidechains. It might be better to show full electron density for a key region of the interface. While the structure is well refined at high resolution, 2Fo-Fc maps using a bias reduced procedure might be helpful to further demonstrate the accuracy of the map.

We thank the reviewer for requesting the improvement. The omit maps displaying the full electron density for the interface are shown as suggested. These modifications are shown in Supplementary Fig. 3.

- It might be better to include the structure overview in Fig. 2, to help orient the reader to understand where in the overall structure the detailed interactions occur.

We agree with the reviewer. We have included a structured overview in Fig.2 and highlight the regions for the zoom in structures for better presentation.

- Structure overview (Fig. 1D) – it would be good to indicate where the binding interface for 53BP1 is located in the green, unbound TIRR protomer. This is important to make the case that two molecules of 53BP1 could in theory bind to a TIRR dimer.

We have modified the structure shown in Fig. 1D and added a sentence to highlight the importance of the availability of the second binding site for Tudor. “It is worth noting that the binding of one 53BP1 Tudor to TIRR dimer does not block the Tudor-binding site on the unbound TIRR protomer, suggesting that two molecules of 53BP1 could in theory bind to one TIRR dimer.” (on page 6, 1st paragraph).

- Table I – Rmerge is 0 in the highest resolution bin? This must be a typo.

Previously we used HKL2000 to generate the structural data in Table 1. It is true that the R_{merge} value in the highest resolution shell is 0.000 although the R_{pim} value of 0.023 (0.645) appear normal (Table from the scaling step of HKL2000, see below). Nonetheless, data procession using CCP4 iMOSFLM resulted in $R_{merge} = 0.107$ (2.258) and $R_{pim} = 0.035$ (0.788), respectively (Table from the scaling step of CCP4 iMOSFLM, see below). It is unclear why data scaling process of HKL2000 produces an abnormal R_{merge} . However, based on these results, it would be fair to say the original data and overall structure are of good quality. This conclusion is further supported by (1) a snapshot of diffraction images (See figure attached) and (2) a PDB structure validation report (See figure attached). Collectively, the $R_{pim} = 0.023$ (0.645) derived from HKL2000 is shown in Table 1 for unambiguous presentation.

Shell limit	Lower Angstrom	Upper Angstrom	Average I	Average error	Average stat.	Norm. Chi**2	Linear R-fac	Square R-fac	Rmeas	Rpim	CC1/2	CC*
50.00	3.79	1607.4	21.5	7.6	1.685	0.047	0.054	0.048	0.011	0.999	1.000	
3.79	3.01	706.4	11.4	4.5	2.005	0.069	0.072	0.071	0.016	0.999	1.000	
3.01	2.63	227.8	3.9	3.0	2.194	0.096	0.095	0.099	0.022	0.998	1.000	
2.63	2.39	124.7	3.2	2.9	1.916	0.143	0.136	0.146	0.032	0.997	0.999	
2.39	2.22	82.3	3.3	3.1	1.778	0.224	0.219	0.230	0.051	0.994	0.999	
2.22	2.09	56.8	3.4	3.4	1.666	0.344	0.339	0.353	0.079	0.989	0.997	
2.09	1.98	32.9	3.5	3.5	1.524	0.600	0.617	0.616	0.139	0.968	0.992	
1.98	1.90	20.2	3.5	3.5	1.432	0.996	0.000	0.000	0.231	0.931	0.982	
1.90	1.82	11.9	3.6	3.6	1.333	0.000	0.000	0.000	0.388	0.825	0.951	
1.82	1.76	7.2	3.7	3.6	1.249	0.000	0.000	0.000	0.645	0.639	0.883	
All reflections		292.4	6.1	3.9	1.688	0.085	0.063	0.081	0.023			

Table from the scaling step of HKL2000

	Overall	InnerShell	OuterShell
Low resolution limit	48.22	48.22	1.80
High resolution limit	1.76	8.64	1.76
Rmerge (within I+/I-)	0.107	0.044	2.258
Rmerge (all I+ and I-)	0.109	0.044	2.324
Rmeas (within I+/I-)	0.112	0.047	2.393
Rmeas (all I+ & I-)	0.111	0.046	2.392
Rpim (within I+/I-)	0.035	0.016	0.788
Rpim (all I+ & I-)	0.025	0.012	0.561
Rmerge in top intensity bin	0.052	-	-
Total number of observations	1421894	10414	80564
Total number unique	73545	639	4548
Mean(I)/sd(I)	18.4	45.1	1.7
Mn(I) half-set correlation CC(1/2)	0.999	0.998	0.603
Completeness	99.9	92.0	100.0
Multiplicity	19.3	16.3	17.7
Anomalous completeness	99.9	89.4	99.7
Anomalous multiplicity	9.9	8.9	9.0
DelAnom correlation between half-sets	-0.157	-0.050	0.005
Mid-Slope of Anom Normal Probability	0.871	-	-
Estimates of resolution limits: overall			
from half-dataset correlation CC(1/2) > 0.50:	limit = 1.76A	== maximum resolution	
from Mn(I/sd) > 2.00:	limit = 1.80A		

Table from the scaling step of CCP4 iMOSFLM

- Some English editing would help with readability of the manuscript.

We thank the reviewer for requesting the improvement. We have done some English editing and will keep improving the manuscript readability if necessary.

Figure. Isothermal titration calorimetric analysis of the binding of TIRR charge reversal mutant to 53BP1 Tudor wt or D1521R mutant. A-B, Titration of TIRR R107D or TIRR R107E mutants with 53BP1 Tudor wt. C-D, Titration of TIRR R107D or TIRR R107E mutants with 53BP1 Tudor D1521R.

Reviewer #2 (Remarks to the Author):

In this manuscript, the authors present the crystal structure of TIRR (NUDT16L1) in complex with the Tudor domain of 53BP1. TIRR was recently discovered as a novel 53BP1-binding protein that regulates its activity in DNA repair. Overall, the structure illuminates the mechanism by which TIRR inhibits 53BP1 and will be of interest to many in the field interested in 53BP1-dependent DNA repair. However, I have a small number of comments/questions that I hope can be answered before publication.

1- While the authors characterize mutations in TIRR that impact its binding to 53BP1, a very useful set of mutations would be mutations, on 53BP1, that selectively disrupt binding to TIRR without impacting binding to H4K20me2. These mutants will be key to understand the role of TIRR in the regulation of 53BP1. From their analyses, Y1523A or Y1523S might be the mutations that most closely correspond to separation-of-function mutations. What is the impact of these mutations on DNA repair? Are they leading to “hyperactive” 53BP1?

We thank the reviewer for the valuable insight into creating a “hyperactive” 53BP1. As suggested by the reviewer, we examined 53BP1 Y1523A, a separation-of-function mutant exclusively influencing TIRR-Tudor interaction, for its effect on IR-induced 53BP1 foci formation and RIF1 (Rap1 interacting factor 1) accumulations at DSB sites. Our results showed in MCF10A derived 53BP1 knockout cells, the 53BP1 D1521R mutation, which disrupts this activity of the Tudor domain, failed to form IR-induced foci whereas 53BP1 Y1523A formed foci that colocalize with γ -H2AX (Supplementary Fig. 5A). It is well known that RIF1 is a downstream effector of 53BP1 during DSB repair, and RIF1 accumulations at DSB sites is dependent on 53BP1 Tudor domain. As shown in Supplementary Fig. 5B, the RIF1 foci is abolished in 53BP1 knockout cells, while RIF1 foci is re-established in the cells overexpressing 53BP1 wild-type or Y1523A mutant, but not the D1521R mutant. Furthermore, foci formation of 53BP1 Y1523A is still observed in cells overexpressing TIRR, however, TIRR expression almost completely abolishes foci formation of 53BP1 wild-type (Supplementary Fig. 5C), indicating that 53BP1 Y1523A mutant is a “hyperactive” 53BP1. These results have been shown in the revised manuscript and discussed accordingly (on page 10, 2nd paragraph).

2- The clear observation that the TIRR KO leads to destabilization of 53BP1 suggests that TIRR might function in the biogenesis of 53BP1. Is there any insight from the structure that could explain the destabilization of 53BP1 in the absence of TIRR?

We had addressed a related issue in response to reviewer 1 by discussing how the binding of 53BP1 Tudor to TIRR may enhance the stability of the complex. To address the issue raised by reviewer 2, we have added one more sentence in the text (on page 13, 3rd paragraph), as follows “We postulated that the robust binding of 53BP1 Tudor to TIRR may block the ubiquitin targeted degradation, which ensues after 53BP1 released from the chromatin epitopes or other binding factors. TIRR therefore can regulate 53BP1 biogenesis by altering the stoichiometry and dynamics of 53BP1 interacting with DSBs”.

Minor comment

- Ref 11 (Huyen et al.) did not show that 53BP1 bound to H4K20me2. It was rather Botuyan et al. (ref 13).

We apologize for the inappropriate citation. The reference citation is changed as suggested.

- I urge the authors to add page numbers.

The page numbers have been added.

REVIEWERS' COMMENTS:

Reviewer #1 (Remarks to the Author):

The authors have done a nice job in responding to my comments.

Reviewer #2 (Remarks to the Author):

In my review, I had suggested that the authors attempt to generate a TIRR-insensitive 53BP1 as a means to understand the consequences of 53BP1 hyperactivation. In the revised manuscript, the authors suggest that the 53BP1 Y1523A mutant is "hyperactive" because it is insensitive to the exogenous expression of TIRR. However, there is no indication that this mutant has augmented 53BP1-dependent DNA repair capacity (e.g. increased capacity at blocking resection). Therefore, it may be misleading to refer to the Y1523A mutant as hyperactive. I simply suggest the authors remove that term (line 278). It seems to me that the authors have missed the opportunity to functionally characterize this mutant.

REVIEWERS' COMMENTS:

Reviewer #1 (Remarks to the Author):

The authors have done a nice job in responding to my comments.

Reviewer #2 (Remarks to the Author):

In my review, I had suggested that the authors attempt to generate a TIRR-insensitive 53BP1 as a means to understand the consequences of 53BP1 hyperactivation. In the revised manuscript, the authors suggest that the 53BP1 Y1523A mutant is "hyperactive" because it is insensitive to the exogenous expression of TIRR. However, there is no indication that this mutant has augmented 53BP1-dependent DNA repair capacity (e.g. increased capacity at blocking resection). Therefore, it may be misleading to refer to the Y1523A mutant as hyperactive. I simply suggest the authors remove that term (line 278). It seems to me that the authors have missed the opportunity to functionally characterize this mutant.

We thank the reviewer for the suggestion. In the revised manuscript, we analyze 53BP1 Y1523A mutant for its effect on IR-induced 53BP1 foci formation and RIF1 accumulations at DSB sites and demonstrated it is insensitive to TIRR regulation. These findings, which are highly consistent with our structure results, further strengthen the revised manuscript. We agree with the reviewer that it is of great interest to characterize the function of 53BP1 Y1523A mutant. However, this study will be more appropriate for future exploration due to the limited time and the scope of the current study. As suggested by the reviewer, we removed the last sentence in line 278 to avoid possible misunderstanding (on page 10, 2nd paragraph).